# JuryProbe: A Consensus-Risk Guardrail for Reference-Free Factuality Judge Panels

## Abstract

Model-based evaluation systems increasingly use panels of inexpensive LLM judges to make accept-or-escalate decisions. In factuality settings, accepting a claim because several reference-free judges agree can create a hidden safety risk: agreement may arise from shared false-negative blind spots rather than independent evidence. We introduce JURYPROBE, a consensus-risk guardrail that decides when reference-free panel accept decisions should be routed to grounded verification. JuryProbe estimates panel-level consensus risk from a calibration probe using false-negative-only (FN-only) judge correlation and false-consensus lift, then protects high-risk accept decisions by escalating them to the same judge panel with reference grounding. The grounded verifier is not a stronger oracle model; it is the same panel given a trusted reference and aggregated by grounded majority. Using frozen FEVER-supported claim pools, fixed-seed construction, and audited Number and Entity corruption families, we show that reference-free judge panels exhibit substantial correlated false negatives. Specifically, FN-only correlation is 0.402 for Number and 0.368 for Entity, with false-consensus lift of 3.13× and 18.13×, respectively. Grounded verification collapses this failure mode, reducing unanimous false consensus to zero in both families. In 10-seed held-out deployment evaluations, JuryProbe-Routed reduces false accepts to 0.000 while using 49.6% fewer verifier calls than Always Grounded for Number and 62.1% fewer for Entity. Disagreement-based and budget-matched random routing fail to eliminate false accepts, showing that JuryProbe's benefit comes from routing on measured consensus risk rather than generic escalation.

## 1 Introduction

In many model-evaluation pipelines, the most consequential action is not producing a score, but deciding whether an output should be accepted without further verification. A cheap reference-free judge panel makes this decision attractive: several models can be queried quickly, and majority or unanimous agreement appears to provide redundancy. Yet this deployment logic relies on an assumption that is rarely tested: agreement is informative only when judge errors are sufficiently independent. If the judges share the same false-negative blind spots, the panel may accept a corrupted claim precisely when it appears most confident. In this setting, agreement is not merely noisy evidence; it can become the event that triggers unsafe acceptance.

This paper studies that failure mode as a guardrail problem. We introduce JURYPROBE, a consensus-risk guardrail for reference-free factuality judge panels. Prior work has established LLM-as-a-judge evaluation as a practical paradigm (Zheng et al., 2023; Wang et al., 2024; Zhu et al., 2025), and recent work has explored replacing single judges with panels of smaller models (Verga et al., 2024). JuryProbe asks a narrower deployment question: when should agreement from a cheap reference-free factuality panel be trusted, and when should it be routed to grounded verification?

JuryProbe estimates consensus risk from a calibration probe using two complementary panel-level statistics. FN-only correlation measures dependence: whether judges fail on the same corrupted claims. False-consensus lift measures consequence: whether unanimous false acceptance occurs more often than expected from the

judges' marginal false-negative rates. Together, these statistics define a panel-level risk regime rather than a sample-level classifier.

This framing differs from uncertainty or disagreement-based escalation. Selective prediction and escalation methods route uncertain cases to stronger systems or humans (Geifman & El-Yaniv, 2017; 2019; Jung et al., 2025). Those methods are valuable, but false consensus is different: the dangerous cases are precisely those in which the reference-free judges agree. A disagreement-based guardrail cannot catch unanimous false acceptance by construction. JuryProbe therefore routes accept decisions when calibration shows that the panel is in a high-risk consensus regime, not merely when the judges disagree on a particular item.

We evaluate JuryProbe on frozen FEVER-supported claim pools with audited Number and Entity corruption families. Reference-free judge panels exhibit substantial consensus risk, while grounded verification collapses the corresponding false-consensus pattern. Held-out guardrail evaluation further shows that JuryProbe-Routed eliminates false accepts while using substantially fewer verifier calls than Always Grounded.

**This paper makes three contributions.**

- We define *consensus risk* for reference-free factuality judge panels and show why FN-only correlation and false-consensus lift are complementary panel-level signals that capture dependence and its deployment consequence.

- We provide mechanism validation showing that consensus risk emerges in reference-free judging and collapses under reference grounding, showing that the reference-augmented protocol removes the observed false-consensus pattern in the evaluated families.

- We evaluate JuryProbe-Routed, a held-out guardrail policy that routes high-risk reference-free accept decisions to grounded verification and compare it against disagreement-based and budget-matched routing baselines.

Together, these results suggest that reliability in factuality judging is better improved by risk-aware grounding than by relying on reference-free agreement or simply adding more reference-free judges.

## 2 Related Work

### 2.1 LLM-as-a-Judge and Judge Panel

LLM-as-a-judge systems have become a practical alternative to human evaluation for open-ended model outputs. Zheng et al. (2023) introduce MT-Bench and Chatbot Arena as scalable evaluation setting based on LLM judgments, while Wang et al. (2024) and Zhu et al. (2025) study automatic and fine-tuned LLM evaluators. Verga et al. (2024) further motivate panels of smaller, diverse judges as an alternative to a single expensive judge. JuryProbe builds on this deployment setting, but studies a different question: not whether judge panels are useful on average, but when reference-free panel agreement should no longer be treated as reliable evidence because judge errors may be correlated on the same factual corruptions.

### 2.2 Judge Correlation and Dependence

The value of a judge panel depends on whether additional judges provide additional independent evidence. Kohli (2026) studies this issue directly, showing that nominally large LLM judge panels can yield far fewer effective independent votes when judge errors are correlated. This motivates JuryProbe: if judges fail on the same examples, then majority or unanimous agreement cannot be interpreted as independent confirmation.

Our contribution is complementary. Kohli (2026) studies judge independence and effective votes in LLM evaluation panels, whereas JuryProbe studies reference-free factuality as a guardrail problem. We narrow the failure mode to correlated false negatives on corrupted factual claims, measure the downstream consequence through false-consensus lift, test grounding as a mechanism-level intervention, and evaluate a routed policy that sends high-risk accept decisions to grounded verification. JuryProbe therefore treats correlated failures

as an operational risk signal that drives verification decisions, rather than solely as a property of panel composition. Put differently, Kohli asks whether additional judges provide independent votes; JuryProbe asks whether observed agreement should be trusted without grounding.

### 2.3 Factuality Verification and Grounding

Factuality evaluation has been studied through atomic fact decomposition, hallucination detection, sampling-based consistency, and retrieval-augmented verification (Min et al., 2023; Wei et al., 2024; Manakul et al., 2023; Li et al., 2023).

JuryProbe builds on the observation that grounding can improve factuality judgments, but it does not propose a new factuality verifier. Instead, grounding is used as an intervention and an escalation target. The question is not whether grounded verification is useful in general, but when a cheap reference-free panel should be routed to it. This distinction is important for deployment: always grounding every claim may be reliable but expensive, whereas relying only on reference-free agreement can be unsafe when judges exhibit correlated false-negative failures.

### 2.4 Selective Evaluation and Escalation

Selective prediction studies how models can abstain or defer on unreliable inputs, trading coverage for reliability (Geifman & El-Yaniv, 2017; 2019). In LLM evaluation, Jung et al. (2025) study escalation based on estimated agreement with human judgment. JuryProbe likewise treats evaluation as a decision problem, but differs in the routing signal: it routes based on measured consensus risk rather than uncertainty, confidence, or disagreement. This distinction matters because disagreement-based escalation cannot catch unanimous false acceptance by construction.

Unlike prior work, JuryProbe combines judge panels, factuality verification, grounded escalation, and consensus-risk-aware routing in a single guardrail framework. Its goal is not to improve factuality verification itself, but to determine when reference-free agreement should no longer be trusted without grounding. A compact comparison with related work is provided in Appendix A.

## 3 JuryProbe Framework

JuryProbe is a guardrail for reference-free factuality judging. Its goal is not to replace a judge panel with a new factuality verifier, but to decide when agreement from a reference-free panel should no longer be treated as sufficient evidence for accepting a claim. The framework consists of four components: a factuality decision setup, a panel-level consensus-risk estimator, a grounded verification component, and a routed accept-protection policy.

### 3.1 Problem Setup

We consider a factuality decision setting in which a claim must be either accepted as factual or rejected as non-factual. Each example consists of a claim $c_i$ and a ground-truth factuality label $y_i \in \{0, 1\}$, where $y_i = 1$ denotes a factual claim and $y_i = 0$ denotes a corrupted claim. When grounded verification is used, a trusted reference $r_i$ is additionally provided.

A judge receives a claim and returns a binary decision. For a panel of $m$ judges, let $z_{ij}^{\mathrm{RF}} \in \{0, 1\}$ denote whether judge $j$ accepts item $i$ in the reference-free setting, where 1 means accept and 0 means reject. Throughout this work, an accept decision means that the judge treats the claim as factual. The panel majority decision is

$$A_i^{\mathrm{RF}} = \mathbb{I}\left[\sum_{j=1}^{m} z_{ij}^{\mathrm{RF}} \geq \left\lceil \frac{m}{2} \right\rceil\right]. \tag{1}$$

In our main experiments, $m = 3$, so majority acceptance means that at least two judges accept the claim.

The critical failure mode is false acceptance of corrupted claims. For a corrupted item, we use the term false negative in the corruption-detection sense: the judge fails to detect the corruption and incorrectly accepts the claim as factual. Thus, for a corrupted item ($y_i = 0$), judge $j$ makes a false negative when $z_{ij}^{\mathrm{RF}} = 1$. A false-consensus event occurs when all judges accept the same corrupted claim:

$$C_i^{\mathrm{RF}} = \mathbb{I}\left[y_i = 0 \ \wedge \ \sum_{j=1}^{m} z_{ij}^{\mathrm{RF}} = m\right]. \tag{2}$$

This event is especially important for deployment because it cannot be detected by disagreement-based escalation. The panel appears maximally reliable, yet all judges have made the same false-accept error. Under independent false-negative errors, the expected frequency of such events is determined by the judges' marginal false-negative rates and should typically be small. JuryProbe is motivated by the observation that false consensus may occur substantially more often than independence would predict.

This setup defines the object of interest: not merely whether individual judges make errors, but whether a judge panel produces correlated false negatives that lead to false consensus.

### 3.2 Consensus Risk

JuryProbe estimates consensus risk on a calibration probe before applying any routed policy to deployment examples. The calibration probe consists of corrupted claims from a fixed corruption family and the reference-free decisions of a judge panel on those claims. The resulting risk estimate is a panel-level statistic: it characterizes whether a particular judge panel tends to share false-negative failures on a class of factual corruptions. It is not a classifier that predicts whether an individual deployment item is safe. Let $\mathcal{N}_{\mathrm{cal}} = \{i : y_i = 0\}$ denote the corrupted items in the calibration probe. For each corrupted item $i \in \mathcal{N}_{\mathrm{cal}}$ and judge $j$, define the false-negative indicator $e_{ij} = \mathbb{I}\left[z_{ij}^{\mathrm{RF}} = 1\right]$, where $z_{ij}^{\mathrm{RF}} = 1$ means that judge $j$ accepts the corrupted claim as factual. The vector $e_{\cdot j}$ therefore records the false-negative pattern of judge $j$ across corrupted calibration items.

JuryProbe uses two complementary statistics. First, FN-only correlation measures whether judges fail on the same corrupted items. For each judge pair $(j, k)$, we compute the Pearson correlation between their false-negative vectors: $\rho_{jk} = \mathrm{corr}\left(e_{\cdot j}, e_{\cdot k}\right)$. Because $e_{ij}$ is binary, Pearson correlation is equivalent to the phi coefficient and provides a standard measure of association between judge error patterns. The panel-level FN-only correlation is the average pairwise correlation,

$$\rho_{\mathrm{FN}} = \frac{2}{m(m-1)} \sum_{1 \leq j < k \leq m} \rho_{jk}. \tag{3}$$

This statistic measures dependence. If judges make false-negative errors independently, then knowing that one judge missed a corruption should provide little information about whether another judge missed the same corruption. Positive FN-only correlation indicates correlated false-negative failures.

Second, false-consensus lift measures the consequence of that dependence for unanimous false acceptance. Let the observed false-consensus rate be

$$q_{\mathrm{obs}} = \frac{1}{|\mathcal{N}_{\mathrm{cal}}|} \sum_{i \in \mathcal{N}_{\mathrm{cal}}} \mathbb{I}\left[\sum_{j=1}^{m} e_{ij} = m\right].$$

Let each judge's marginal false-negative rate be

$$p_j = \frac{1}{|\mathcal{N}_{\mathrm{cal}}|} \sum_{i \in \mathcal{N}_{\mathrm{cal}}} e_{ij}.$$

Under independent false-negative errors with the same marginal rates, the expected unanimous false-consensus rate is $q_{\mathrm{ind}} = \prod_{j=1}^{m} p_j$. False-consensus lift is then

$$L_{\text{FC}} = \frac{q_{\text{obs}}}{q_{\text{ind}}} \tag{4}$$

Lift measures consequence rather than dependence itself. A panel can have correlated false negatives, but the operational risk depends on whether this dependence produces more unanimous false acceptance than expected under independence. Values above 1 indicate excess false consensus relative to the independent-error baseline.

JuryProbe uses both statistics because they capture complementary aspects of consensus risk. FN-only correlation measures dependence among judge failures, while false-consensus lift measures the deployment consequence of that dependence. Together, they define a consensus-risk regime for a judge panel. Importantly, consensus risk is assessed at the panel level and remains fixed throughout deployment; it is not recomputed for individual claims.

To assess whether the observed unanimous false-consensus rate is stronger than expected from marginal false-negative rates alone, JuryProbe uses a permutation test with 3000 permutations. The test preserves each judge's marginal false-negative rate while shuffling the locations of false negatives across calibration items. This produces a null distribution in which judges have the same individual error rates but no item-level alignment. The $p$-value is computed as the fraction of shuffled panels whose unanimous false-consensus rate is at least as large as the observed rate, with one-count smoothing.

A panel is treated as high-risk when the calibration probe satisfies three conditions:

$$\rho_{\text{FN}} > 0.15, \quad L_{\text{FC}} > 1.5, \quad p < 0.05.$$

These thresholds define a risk regime, not an item-level decision rule. They were selected prior to deployment evaluation and are examined through a threshold-sensitivity analysis in Section 5.4. The risk regime is estimated only on calibration data and then carried forward to deployment evaluation. In the JuryProbe policy, the high-risk label means that reference-free agreement should no longer be treated as sufficient evidence for accepting claims without grounding.

### 3.3 Grounded Verification

Grounded verification defines the reference-augmented protocol used for mechanism validation and routed escalation. The same judge panel evaluates the same claim with access to a trusted reference; judge identities, binary output space, and majority aggregation are kept fixed.

We denote reference-free decisions by $z_{ij}^{\text{RF}}$ and grounded decisions by $z_{ij}^{\text{G}}$. In the reference-free setting, judge $j$ receives only the claim $c_i$ and returns $z_{ij}^{\text{RF}} \in \{0,1\}$. In the grounded setting, the same judge receives $(c_i, r_i)$, where $r_i$ is the trusted reference, and returns $z_{ij}^{\text{G}} \in \{0,1\}$. The ground-truth label $y_i$ is not provided to the grounded verifier.

Grounded majority acceptance is defined analogously to Eq. 1, replacing $z_{ij}^{\text{RF}}$ with $z_{ij}^{\text{G}}$:

$$A_i^{\text{G}} = \mathbb{I}\left[\sum_{j=1}^{m} z_{ij}^{\text{G}} \geq \left\lceil \frac{m}{2} \right\rceil \right]. \tag{5}$$

For corrupted items, grounded false consensus is defined analogously to Eq. 2, replacing $z_{ij}^{\text{RF}}$ with $z_{ij}^{\text{G}}$:

$$C_i^{\text{G}} = \mathbb{I}\left[ y_i = 0 \ \wedge \ \sum_{j=1}^{m} z_{ij}^{\text{G}} = m \right]. \tag{6}$$

We use the term *grounding collapse* to describe, when observed, an empirical reduction in false-consensus events when moving from $C_i^{\text{RF}}$ to $C_i^{\text{G}}$ while holding the judge panel fixed. Section 5.2 evaluates whether such a collapse occurs using paired reference-free and grounded decisions.

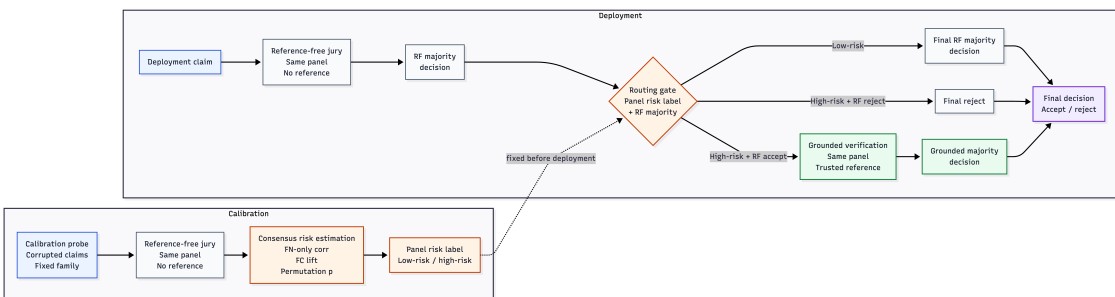

Figure 1: JuryProbe-routed policy. Consensus risk is estimated once on a calibration probe and fixed during deployment. If the panel is high-risk, reference-free majority accept decisions are routed to grounded verification using the same judge panel with trusted references; reference-free rejects are kept as rejects.

### 3.4 JuryProbe-Routed Policy

JuryProbe uses the consensus-risk estimate to decide whether reference-free accept decisions require grounded verification. Let $h \in \{0, 1\}$ denote the panel-level risk label estimated on the calibration probe, where $h = 1$ indicates that the panel is high-risk according to the thresholds in Section 3.2. This risk label is fixed before deployment evaluation. It is not recomputed for each deployment item.

For a deployment claim $i$, the reference-free panel first produces the majority decision $A_i^{\mathrm{RF}}$ defined in Eq. 1. If the panel is low-risk, JuryProbe uses the reference-free majority decision directly. If the panel is high-risk, JuryProbe routes only reference-free majority accept decisions to grounded verification. Reference-free majority rejects are kept as rejects. The final JuryProbe decision is

$$
D_i^{\mathrm{JP}} = \begin{cases} A_i^{\mathrm{RF}}, & h = 0, \\ A_i^{\mathrm{G}}, & h = 1 \ \wedge \ A_i^{\mathrm{RF}} = 1, \\ 0, & h = 1 \ \wedge \ A_i^{\mathrm{RF}} = 0. \end{cases} \tag{7}
$$

Figure 1 summarizes the routed policy.

This asymmetric design is intentional. JuryProbe is optimized to prevent unsafe acceptance of corrupted claims rather than to maximize acceptance coverage. In factuality verification, accepting a corrupted claim is the safety-relevant error: it allows a false statement to pass as factual. By contrast, a reference-free reject already blocks the claim. JuryProbe therefore does not use grounded verification to recover additional accepts; it uses grounding to prevent high-risk false accepts.

The policy also differs from disagreement-based escalation: JuryProbe targets cases where agreement itself is risky. If calibration shows that a high-risk panel can produce correlated false negatives, even a unanimous reference-free accept can be routed to grounded verification.

Verifier usage is reported using the metrics in Section 4.3; evaluation baselines are defined in Section 4.4.

## 4 Experimental Setup

We evaluate JuryProbe in a controlled factuality setting where clean claims are supported by trusted references and corrupted claims have known non-factual labels. This section describes the frozen corruption datasets, judge panels, metrics, and held-out protocol used to separate risk estimation from policy evaluation.

### 4.1 Claim Pools and Corruption Families

We construct examples from FEVER-supported claims (Thorne et al., 2018). To avoid adaptive example selection, claim pools, corruption generation, and audits are frozen before judge evaluation.

The confirmatory evaluation uses two audited corruption families. Number corruptions modify numerical facts such as years, counts, or quantities, while Entity corruptions replace named entities with plausible alternatives. Each confirmatory family contains 300 clean and 300 corrupted examples, supporting equal-sized calibration and deployment splits. This evaluation budget was fixed before judge evaluation.

We additionally construct an Attribute family for replication analysis. Relation corruptions were explored but excluded before judge evaluation because audits revealed unstable fluency, syntax, and semantic-naturalness artifacts.

## 4.2 Judge Panels

Our main judge panel consists of three relatively small LLM judges: Llama-3.1-8B-Instruct, Qwen-2.5-7B-Instruct, and Gemma-3-12B-IT. The same three judges are used for both reference-free judging and grounded verification. In the reference-free setting, judges receive only the claim; in the grounded setting, they receive the same claim with a trusted reference and determine whether the claim is fully consistent with that reference. Holding judge identities fixed keeps judge composition constant while the protocol changes by adding trusted references. All outputs are converted into binary accept/reject decisions.

## 4.3 Metrics

We report metrics for three purposes: consensus-risk estimation, mechanism validation, and guardrail evaluation.

For consensus-risk estimation, we report FN-only correlation ($\rho_{\mathrm{FN}}$), false-consensus lift ($L_{\mathrm{FC}}$), and the permutation-test $p$-value defined in Section 3.2.

For mechanism validation, we compare reference-free and grounded judging using FN-only correlation and unanimous false-consensus rate.

For guardrail evaluation, we report false accept rate, true accept rate, residual false-consensus rate, extra verifier items, and extra verifier calls. False accept rate is the fraction of corrupted claims accepted by the final policy. True accept rate is the fraction of clean claims accepted by the final policy. Residual false-consensus rate is the fraction of corrupted claims unanimously accepted by the reference-free panel and not blocked by the final policy. Extra verifier items denote the number of deployment examples routed to grounded verification, and extra verifier calls denote the corresponding number of grounded judge calls.

## 4.4 Held-out Evaluation Protocol

For guardrail evaluation, we separate risk estimation from policy evaluation using held-out splits. For each corruption family and split seed, the 300 clean and 300 corrupted examples are partitioned into equal-sized calibration and deployment splits, each containing 150 clean claims and 150 corrupted claims. The calibration split is used only to estimate the panel-level risk label $h$ using the thresholds in Section 3.2. The resulting risk label is fixed before deployment evaluation and is not recomputed on deployment examples.

We repeat this procedure over 10 split seeds and report mean and standard deviation across splits. This multi-seed protocol tests whether the high-risk label and policy outcomes are stable under different calibration/deployment partitions.

We compare JuryProbe-Routed against five baselines: Reference-Free Majority, Reference-Free Unanimity, Always Grounded, Disagreement-Routed, and Random-Routed. Always Grounded sends all deployment claims to grounded verification and serves as a fully grounded comparison policy. Disagreement-Routed escalates only when reference-free judges disagree. Random-Routed uses the same grounded-verification budget as JuryProbe-Routed but selects deployment items uniformly at random from the deployment split, isolating the effect of consensus-risk routing from simply spending additional verifier calls.

All policies are evaluated on the same deployment splits using the metrics in Section 4.3. Missing grounded decisions, if any, are treated as missing rather than replaced with gold labels, ensuring that evaluation reflects actual verifier outputs rather than oracle fallback.

# 5 Results

We organize the results into four parts: consensus-risk estimation, grounding-based mechanism validation, guardrail evaluation, and robustness analyses.

## 5.1 Consensus Risk in Reference-Free Judging

We first test whether reference-free judge panels exhibit consensus risk before any grounded intervention is applied. Table 1 reports the calibration statistics for the audited corruption families. FN-only correlation is computed over all corrupted claims. False-consensus rate, false-consensus lift, and permutation $p$-value are reported on a detectable corrupted subset. A corrupted claim is considered detectable if an external high-capability factuality detector (GPT-4o) correctly rejects the corruption. This subset reduces the influence of claims that may be ambiguous or externally difficult to detect, while preserving the false-consensus events used to evaluate correlated false acceptance.

Table 1: Reference-free consensus-risk statistics. $N_{corr}$ is the number of corrupted claims, $N_{det}$ is the detectable corrupted subset used for difficulty-controlled false-consensus analysis, Corr. denotes FN-only correlation, and FC denotes false consensus. All three families satisfy the high-risk criteria selected prior to deployment evaluation: $\rho_{FN} > 0.15$, $L_{FC} > 1.5$, and permutation $p < 0.05$.

| Family | $N_{corr}$ | $N_{det}$ | Corr. | FC Rate | FC Lift | $p$ |
|---|---|---|---|---|---|---|
| Number | 300 | 214 | 0.402 | 0.159 | 3.13× | < 0.001 |
| Entity | 300 | 286 | 0.368 | 0.031 | 18.13× | < 0.001 |
| Attribute | 300 | 296 | 0.263 | 0.003 | 54.55× | 0.018 |

The two confirmatory families both show substantial consensus risk. Number corruptions produce high FN-only correlation ($\rho_{FN} = 0.402$) and a false-consensus lift of 3.13×. Entity corruptions show a similar dependence pattern ($\rho_{FN} = 0.368$) but an even larger lift of 18.13×, indicating that unanimous false acceptance occurs far more often than expected from marginal false-negative rates alone.

Attribute provides an additional replication of the consensus-risk signal. It also satisfies the high-risk criteria, with $\rho_{FN} = 0.263$, false-consensus lift of 54.55×, and $p = 0.018$. However, its absolute false-consensus rate is much lower (0.003). This low event rate limits the usefulness of Attribute for grounding-collapse analysis, but it still provides an independent replication of the consensus-risk signal beyond Number and Entity.

Together, these results show that reference-free panel agreement can reflect correlated false-negative failures rather than independent confirmation, even when multiple judges agree. This motivates the next question: whether providing trusted references to the same judges collapses the false-consensus failure mode.

## 5.2 Grounding Collapse as Mechanism Validation

We next test whether the consensus risk observed in reference-free judging collapses when the same judges receive trusted references. This analysis uses the two confirmatory families, Number and Entity; Attribute is reported as replication evidence but is not used for grounding-collapse analysis, because its absolute false-consensus event rate is too low to provide a stable paired comparison.

Table 2 compares reference-free and grounded judging on the same GPT-4o detectable corrupted claims. The reference-free statistics in this table are computed on that subset and therefore differ slightly from the all-corrupted FN-only correlation reported in Table 1. The judge identities and aggregation rule are held fixed; while the protocol changes by adding trusted references.

For both confirmatory families, grounding removes the false-consensus pattern. Number falls from RF correlation 0.386 and false-consensus rate 0.159 to grounded correlation 0.000 and false-consensus rate 0.000. Entity shows the same pattern, with RF correlation 0.393 and false-consensus rate 0.031 falling to approximately zero under grounding.

Table 2: Grounding collapse on the confirmatory families. RF denotes reference-free judging, Corr. denotes FN-only correlation, and FC denotes unanimous false consensus. The same judge panel evaluates the same corrupted claims with and without trusted references. Grounded FC lift is not reported because, in both families, the grounded false-consensus rate and the corresponding independence baseline are zero, making the lift ratio undefined.

| Family | RF Corr. | Grounded Corr. | RF FC Rate | Grounded FC Rate | RF FC Lift |
|--------|----------|----------------|------------|------------------|------------|
| Number | 0.386 | 0.000 | 0.159 | 0.000 | 3.13× |
| Entity | 0.393 | -0.003 | 0.031 | 0.000 | 18.13× |

This provides mechanism-level evidence for JuryProbe's diagnosis: the same judge panels exhibit correlated false consensus without references, but the pattern collapses under the reference-augmented protocol. Grounding therefore acts as an intervention on the reference-free failure condition rather than merely as a stronger downstream classifier.

### 5.3 Guardrail Evaluation Against Routing Baselines

We next evaluate whether consensus-risk routing improves deployment decisions. For each family, risk is estimated only on the calibration split and policy metrics are reported on held-out deployment examples over 10 split seeds. In both Number and Entity, JuryProbe detects a high-risk panel in all 10 splits.

Table 3 compares JuryProbe-Routed with reference-free, grounded, disagreement-based, and budget-matched routing baselines. JuryProbe-Routed eliminates false accepts and residual false consensus in both confirmatory families. For Number, false accept rate falls from 0.427 under Reference-Free Majority to 0.000 under JuryProbe-Routed. For Entity, false accept rate falls from 0.119 to 0.000. True accept rate is preserved relative to Reference-Free Majority because JuryProbe only routes reference-free accept decisions and keeps reference-free rejects unchanged.

Table 3: Held-out guardrail evaluation over 10 split seeds. Values are mean ± standard deviation across splits. Residual FC denotes residual false-consensus rate after policy decisions. Verifier Calls denotes the number of grounded judge calls used by the policy. Always Grounded uses actual grounded verifier outputs from the same judge panel, not oracle labels.

| Family | Policy | False Accept | True Accept | Residual FC | Verifier Calls |
|--------|--------|--------------|-------------|-------------|----------------|
| Number | RF Majority | 0.427±0.029 | 0.581±0.029 | 0.201±0.022 | 0±0 |
| Number | RF Unanimity | 0.201±0.022 | 0.273±0.027 | 0.201±0.022 | 0±0 |
| Number | Disagreement-Routed | 0.201±0.022 | 0.762±0.029 | 0.201±0.022 | 420±19 |
| Number | Random-Routed | 0.212±0.013 | 0.792±0.019 | 0.100±0.009 | 454±14 |
| Number | **JuryProbe-Routed** | **0.000±0.000** | 0.581±0.029 | **0.000±0.000** | 454±14 |
| Number | Always Grounded | 0.000±0.000 | 1.000±0.000 | 0.000±0.000 | 900±0 |
| Entity | RF Majority | 0.119±0.014 | 0.639±0.027 | 0.035±0.011 | 0±0 |
| Entity | RF Unanimity | 0.035±0.011 | 0.394±0.027 | 0.035±0.011 | 0±0 |
| Entity | Disagreement-Routed | 0.035±0.011 | 0.841±0.015 | 0.035±0.011 | 306±10 |
| Entity | Random-Routed | 0.074±0.008 | 0.776±0.021 | 0.022±0.007 | 341±13 |
| Entity | **JuryProbe-Routed** | **0.000±0.000** | 0.639±0.027 | **0.000±0.000** | 341±13 |
| Entity | Always Grounded | 0.000±0.000 | 1.000±0.000 | 0.000±0.000 | 900±0 |

Disagreement-Routed leaves residual false consensus because unanimous false accepts provide no disagreement signal. Random-Routed fails to eliminate false accepts despite using the same verifier-call budget as JuryProbe-Routed. Always Grounded also eliminates false accepts, but requires grounded verification for every deployment item. JuryProbe-Routed matches this false-accept reduction while using 49.6% fewer grounded verifier calls for Number and 62.1% fewer for Entity.

These results show that the improvement comes from routing on measured consensus risk rather than from generic escalation or additional verifier usage.

### 5.4 Robustness and Replication

We perform four additional analyses to evaluate whether the main findings are sensitive to corruption family, judging condition, threshold choice, or judge-panel composition. These analyses support the same interpretation: the observed consensus-risk pattern is not driven by a single corruption family, a grounded-estimator artifact, a narrow threshold choice, or one particular judge-panel composition. Detailed robustness results are reported in Appendix B.

## 6 Discussion

JuryProbe treats reference-free agreement as reliable only after the dependence structure of judge errors has been characterized. The remainder of this section discusses the limitations of the current study and directions for future work.

### 6.1 Limitations

This study focuses on controlled factuality judgments over audited corruption families. The main mechanism-validation and guardrail-policy results use Number and Entity corruptions, while Attribute is reported as replication evidence. Although these families cover distinct factual substitution types, they do not exhaust the space of factual errors. Consensus-risk estimates are therefore family-dependent and may not transfer unchanged to unseen corruption types. Relation corruptions were explored during construction but excluded because audits revealed unstable fluency, syntax, and semantic naturalness artifacts.

The detectable-subset analysis relies on GPT-4o as an external analysis-time detector. Although this detector is not part of the JuryProbe panel, grounded verifier, or routed policy, its use remains a modeling choice. Appendix D.4 reports all-corrupted and grounded-detectable variants to check that the qualitative conclusions do not depend on this choice.

The current framework assumes that trusted references are available and sufficiently accurate for grounded verification. The interaction between consensus-risk routing and noisy retrieval, incomplete evidence, or incorrect references is not examined in this work.

Grounding collapse is evaluated by comparing paired reference-free and grounded decisions while holding the judge panel fixed. Because grounded verification changes the input protocol by introducing trusted references, this comparison provides mechanism validation for the reference-augmented protocol rather than a complete causal isolation of reference wording, prompt framing, or other protocol-level effects.

Finally, JuryProbe is designed as an accept-protection guardrail. The routed policy protects against unsafe acceptance of corrupted claims but does not attempt to recover additional true accepts from reference-free rejects. Although the detectable-subset analysis reduces the influence of shared difficulty, it cannot completely eliminate all latent sources of item difficulty, a concern also raised in studies of correlated judge errors (Kohli, 2026).

### 6.2 Future Work

Extending this idea beyond factuality remains an important direction for future work. Many emerging uses of LLM judges involve planning, forecasting, recommendation, and other settings in which trusted references may be incomplete or unavailable. Developing risk-aware routing and intervention strategies for these evidence-sparse settings remains an open challenge, but the central lesson remains the same: agreement should not be treated as reliability unless the dependence structure of judge errors is understood.

## 7 Conclusion

This paper introduced JuryProbe, a consensus-risk guardrail for reference-free factuality judge panels. Rather than treating agreement as inherently reliable, JuryProbe estimates whether a judge panel exhibits correlated false-negative failures that can produce false consensus. The framework combines panel-level risk estimation

using FN-only correlation and false-consensus lift, mechanism validation through grounding collapse, and a routed policy that sends high-risk accept decisions to grounded verification.

Across audited Number and Entity corruption families, we find that reference-free judge panels exhibit substantial correlated false negatives and excess unanimous false acceptance. When the same judges are provided with trusted references, this failure mode collapses, supporting the interpretation that consensus risk arises from judging factuality without sufficient evidence rather than from judge identity alone. In held-out deployment evaluations, JuryProbe-Routed eliminates false accepts and residual false consensus while using substantially fewer grounded verifier calls than Always Grounded.

Taken together, these results suggest that reliability in LLM factuality judging depends not only on individual judge accuracy or panel agreement, but also on the dependence structure of judge errors. Agreement should not be treated as sufficient evidence for factual acceptance unless the dependence structure underlying that agreement is understood.

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

## A    Related Work Positioning

Table 4: Positioning of JuryProbe relative to related work. Panel = LLM judge panel; Fact. = factuality-oriented evaluation; Grounded = grounded verification; Escal. = selective escalation; Consensus Risk = routing based on measured correlated-failure risk.

| Work | Panel | Fact. | Grounded | Escal. | Consensus Risk |
|---|---|---|---|---|---|
| Kohli (2026) | ✓ | x | x | x | Partial |
| Trust or Escalate | ✓ | x | Partial | ✓ | x |
| PoLL | ✓ | x | x | x | x |
| LongFact | x | ✓ | x | x | x |
| SAFE | x | ✓ | ✓ | x | x |
| FActScore | x | ✓ | ✓ | x | x |
| HaluEval | x | ✓ | x | x | x |
| **JuryProbe** | ✓ | ✓ | ✓ | ✓ | ✓ |

## B    Additional Robustness and Artifact Details

Table 5 reports the robustness checks summarized in Section 5.4. These checks test whether the main conclusions depend on threshold choice, grounded-output specificity, random-routing variance, grounded evaluation integrity, or judge-panel composition. All checks are computed from frozen datasets and cached model outputs.

## C    Representative False-Consensus Cases

Table 6 shows representative false-consensus cases. In each case, all three reference-free judges accept the corrupted claim, even though the trusted reference contradicts the modified factual element. When the same judges receive the reference, grounded verification rejects the claim. These examples illustrate why disagreement-based routing is insufficient: the reference-free panel does not disagree on these corrupted claims; it unanimously accepts them.

Table 5: Additional robustness and artifact checks. Threshold sensitivity reports high-risk detections over correlation thresholds from 0.10 to 0.25 and lift thresholds from 1.25 to 2.00. Grounded specificity reports high-risk detections when the same risk estimator is applied to grounded verifier outputs.

| Check | Result | Purpose |
|---|---|---|
| Threshold sensitivity | Number: 10/10; Entity: 10/10 | Not threshold-fragile |
| Grounded specificity | Number: 0/10; Entity: 0/10 | Not always high-risk |
| Random-Routed stability | 100 trials per split | Stable budget-matched baseline |
| Grounded evaluation integrity | No oracle, no gold fallback, no missing imputation | Actual verifier outputs only |
| Strong judge slice | $\rho_{\text{FN}} = 0.252$, $L_{\text{FC}} = 2.17\times$, $p < 0.001$ | Stronger RF judges do not remove risk |

Table 6: Representative false-consensus cases. RF denotes reference-free judging, and G denotes grounded judging with the same judge panel and trusted reference.

| Family | Corrupted claim | Reference | Outcome |
|---|---|---|---|
| Number | Pacific Rim was released July 14, 2013. | Pacific Rim was released July 12, 2013. | RF: 3/3 accept; G: 0/3 accept |
| Number | Alex Rodriguez was suspended for 169 games. | Alex Rodriguez was suspended for 211 games. | RF: 3/3 accept; G: 0/3 accept |
| Entity | Steve Jobs's birth date is October 28, 1955. | Bill Gates's birth date is October 28, 1955. | RF: 3/3 accept; G: 0/3 accept |
| Entity | Elton John was named MusiCares' person of the year in 2013. | Bruce Springsteen was named MusiCares' person of the year in 2013. | RF: 3/3 accept; G: 0/3 accept |

## D Additional Experimental Details

### D.1 Judge Prompts and Output Parsing

All reference-free and grounded judgments are converted into binary decisions before evaluation. In the reference-free condition, each judge receives only the claim and is asked to decide whether the statement is factually correct. The prompt requires a one-word output, `true` if the statement is fully correct and `false` if it contains any factual error. In the grounded condition, each judge receives a trusted reference together with the claim and is asked whether the claim is fully consistent with the reference. The grounded prompt also requires a one-word `true`/`false` output. Outputs are parsed using a strict string matcher for `true` or `false`. Responses that do not contain a valid binary verdict are marked as parse failures rather than interpreted manually. For the final grounded-verifier cache used in the guardrail evaluation, parse failures are retried with the same grounded prompt. The validated cache contains the expected number of grounded decisions for both confirmatory families: 1800 grounded judge outputs for Number and 1800 for Entity, corresponding to 600 examples times three judges. The final validated cache has zero remaining parse failures. No oracle replacement, gold-label fallback, or missing-item imputation is used. Grounded decisions are produced by the same judge panel used in reference-free evaluation, with the only protocol change being the addition of the trusted reference to the judge input. The ground-truth label is never provided to the judge and is used only for evaluation.

### D.2 Corruption Audit Protocol

All corruption families are frozen before judge evaluation. Number and Entity are used as confirmatory families because author audits found them reliable enough for the main consensus-risk, mechanism-validation, and guardrail-policy evaluations. Number corruptions modify numerical facts such as years, counts, or quantities, while Entity corruptions replace named entities with plausible alternatives. Attribute is reported as an additional replication family because it satisfies the consensus-risk criteria but has a very low absolute false-consensus event rate, limiting its usefulness for grounding-collapse and guardrail-policy evaluation.

Relation corruptions were explored during construction but excluded from judge evaluation because audits revealed unstable fluency, syntax, and semantic naturalness artifacts. This exclusion was made before using Relation in any main judge-panel evaluation.

## D.3 Detectable Subset Analysis

False-consensus lift and grounding-collapse analyses are reported on a detectable corrupted subset to reduce the influence of items that may be difficult or ambiguous even for a stronger external factuality detector. Detectability is estimated using GPT-4o as an analysis-time external detector. GPT-4o is not part of the JuryProbe judge panel, is not used as the grounded verifier, and is not used by the routed policy. A corrupted item is included in the detectable subset if GPT-4o correctly rejects the corrupted claim. This analysis-only filter focuses the false-consensus analysis on corrupted claims whose factual error is externally detectable, while still evaluating whether the reference-free judge panel unanimously accepts them. The detectable subset contains 214, 286, and 296 corrupted examples for Number, Entity, and Attribute, respectively. FN-only correlation is reported over all corrupted claims, while false-consensus rate, false-consensus lift, and the permutation-test $p$-value are reported on the detectable corrupted subset. Grounding-collapse analysis uses the same detectable corrupted claims in paired reference-free and grounded conditions.

## D.4 Alternative Detectability Definitions

The main analysis uses the GPT-4o detectable subset because this filter is independent of both the JuryProbe judge panel and the grounded verifier. To check that the conclusions do not depend on this particular filter, we also recompute the reference-free false-consensus statistics under two alternative subsets: all corrupted claims, and a grounded-detectable subset consisting of corrupted claims for which at least one grounded judge rejects the claim. The grounded-detectable variant is reported only as a robustness check, not as the primary mechanism-validation subset, because it is defined using grounded verifier outputs.

Table 7: Alternative detectability definitions. Across all detectability definitions, the qualitative conclusion remains unchanged: reference-free panel exhibits excess false consensus, while grounded false consensus is eliminated.

| Family | Subset | N | RF Corr. | RF FC Rate | RF FC Lift | Grounded FC |
|--------|--------|-----|----------|------------|------------|-------------|
| Number | GPT-4o detectable | 214 | 0.386 | 0.159 | 3.13× | 0.000 |
| Number | All corrupted | 300 | 0.402 | 0.193 | 2.69× | 0.000 |
| Number | Grounded-detectable | 300 | 0.402 | 0.193 | 2.69× | 0.000 |
| Entity | GPT-4o detectable | 286 | 0.393 | 0.031 | 18.13× | 0.000 |
| Entity | All corrupted | 300 | 0.368 | 0.030 | 13.19× | 0.000 |
| Entity | Grounded-detectable | 300 | 0.368 | 0.030 | 13.19× | 0.000 |

