# OpenReview forum: "JuryProbe: A Consensus-Risk Guardrail for Reference-Free Factuality Judge Panels"
_TMLR — Under review for TMLR_

### Review · Reviewer_f9gw · 2026-07-13

**Summary Of Contributions:**

The paper addresses a weakness of using a panel of cheap LLM judges to check facts without a reference: when the judges share the same blind spots, they can all wrongly accept a false claim at once, so their agreement becomes a risk rather than evidence. To catch this, the authors propose JuryProbe, a two-step pipeline. (1) Offline calibration: on known-corrupted claims, they measure how often the judges make the same mistakes (error correlation) and how much more often they all wrongly accept a claim than independent errors would predict (false-consensus lift); if both are high, the panel gets a high-risk label. (2) Deployment: the panel votes with no reference; if it is high-risk, any claim it accepts is re-checked by the same judges with a trusted reference, while rejections are kept.

The contribution is thus to reframe panel agreement as a risk to be measured, and to turn that into a concrete guardrail plus routing rule.

**Strengths**. Clear setup with data splits and multi-seed reporting.

**Weaknesses**. The paper is called a risk guardrail but gives no real guarantee that risk is controlled; its own always-check-with-reference baseline is simpler and actually better; and the risk score ends up doing nothing beyond always re-checking accepted claims.

**Audience:**

Yes

**Audience Explanation:**

Yes. LLM judge panels are widely used, and the core point is that agreement can be the trigger for a mistake, so disagreement-based checks are a clean, useful insight.

**Broader Impact Concerns:**

No major concerns.

**Claims And Evidence:**

No

**Claims Explanation:**

The experiments are run carefully, but the main claims are not convincingly supported. Three points below:

1. The method is claimed as a risk guardrail, but it has no theoretuical gaurantee on risk. This stands out because the paper compares itself to methods that do give theoretical guarantees [1,2] yet offers something weaker. In practice the thresholds (correlation > 0.15, lift > 1.5, p < 0.05) are hand-picked, and there is no way for a user to set a target error rate and get a rule that meets it. The permutation p-value only tests whether false consensus is higher than chance, which says nothing about the actual deployment error the guardrail is supposed to limit.

2. Why not always check with reference baseline? It also simple, no additional risk and better. In Table 3, Always Grounded also reaches 0.000 false accepts and accepts 100% of the true claims, while JuryProbe accepts only 58–64%. Since both methods assume a reference is available anyway, JuryProbe's only advantage is doing fewer reference-checks，but this is reported as a bare ratio, with no cost model, and the checker is just the same three small models. The paper never explains why anyone would pick JuryProbe over simply always using the reference.

3. The risk score never actually changes what happens. The panel is labeled high-risk in every case tested (Number, Entity, Attribute; all 10 splits). So JuryProbe always does the same thing: re-check every accepted claim. There is no example where the panel is low-risk and the method correctly decides not to re-check while staying safe. Without such a case, there is no evidence the risk measurement does anything beyond always re-checking accepts, which is the paper's central claim.

[1] Jung, Jaehun, Faeze Brahman, and Yejin Choi. "Trust or escalate: Llm judges with provable guarantees for human agreement." International Conference on Learning Representations. Vol. 2025. 2025.

[2] Geifman, Yonatan, and Ran El-Yaniv. "Selective classification for deep neural networks." Advances in neural information processing systems 30 (2017).

**Requested Changes:**

The following are fundamental: each targets a reason the central claims are currently unsupported, and I do not believe they can be resolved without substantial new experiments.

1. Provide a provable risk guarantee, or downweight the risk-guardrail framing.

2. Demonstrate that the risk score changes the decision. Provide at least one setting in which the panel is genuinely low-risk and JuryProbe correctly chooses not to re-check while remaining safe. Without such a case, the method is indistinguishable from always re-checking every accepted claim.

3. Justify JuryProbe against the Always-Grounded baseline. Always Grounded is simpler and dominates on both false accepts (0.000) and true accepts (100% vs. 58–64%). Add a concrete cost comparison (latency or dollars per check on these small judges) and identify the regime, if any, where fewer reference-checks outweighs accepting far fewer true claims.

4. Re-ground or re-label the grounding-collapse result. Evaluate with realistic (retrieved or imperfect) references, or clearly present the current numbers as a best case under a perfect reference and drop the collapse-to-zero and mechanism-validation language.

---

> ### Author Response · Authors · 2026-07-15
> **Author Response (1/2): New Negative-Control Experiment and Grounded-Results Relabeling**
>
> Thank you for the careful review. Our full response exceeds the comment length limit, so we post it in two parts. This part covers the new negative-control experiment (Change 2) and the relabeling of the grounded results (Change 4); Part 2 covers the cost analysis (Change 3) and the risk-guarantee framing with our revision plan (Change 1).
>
> **Change 2: The calibration signal changes the decision — new negative control**
>
> We constructed a negative-control family using the same source pool, judges, splits, unchanged thresholds, and evaluation code as the submitted experiments; only the corruption construction differs. **Self-Contained Contradiction Control** (n = 300 clean / 300 corrupted): each corrupted claim appends one of 40 distinct self-contained numeric contradictions (year ordering, magnitude, arithmetic) to a clean claim. This control is intentionally simple: it tests whether the gate can close in a setting without unanimous false consensus, not whether natural deployment environments are safe.
>
> **Results (10 held-out splits):**
>
> | Control | Flagged high-risk splits | FN Corr | All-3 FC | FC Lift | $p$ | RF False Accept | True Accept | Verifier Calls |
> |---|---:|---:|---:|---:|---:|---:|---:|---:|
> | Self-Contained Contradiction | **0/10** | 0.082 ± 0.054 | 0.000 | 0.000 in 9 splits; undef. in 1 | 1.000 (all 10 splits) | 0.013 ± 0.006 | 0.552 ± 0.023 | **0** |
>
> *Implementation note.* In the split containing a constant all-zero false-negative vector for one judge, pairwise FN correlations involving that judge are set to 0.0 under the same zero-variance convention used in the submitted implementation; FC lift is marked undefined because the independent false-consensus baseline is zero.
>
> The control is not flagged as high-risk in any of the 10 splits, so JuryProbe selects the no-routing branch and performs **zero grounded re-checks**. This is consistent with the target failure mode: the control shows no observed unanimous false consensus; in the nine splits where lift is defined, the lift is zero. The deployed reference-free majority-vote false-accept rate is 0.013 ± 0.006, and **no corrupted claim is accepted unanimously in any split** (all-3 false consensus is exactly zero). We will make this scope explicit: not being flagged as high-risk is not a zero-error certificate; it means that the pre-specified calibration criteria for grounded routing were not met. This contrasts with the 0.119–0.427 reference-free false-accept rates observed in the high-risk families. This control addresses the branch-activation concern; it is not presented as evidence of a formal safety guarantee or as an optimal natural-deployment operating point.
>
> Together with the main experiments, where the same rule and unchanged thresholds flag Number and Entity as high-risk in 10/10 splits and trusted-reference re-checking yields zero observed false accepts, the risk score now produces both outcomes across settings, and the induced routing differs accordingly. The new control is a specificity check: it shows that the calibration rule is not merely an always-on trigger for re-checking accepted claims.
>
> **Change 4: Grounded results — relabeled**
>
> The grounded experiments used a trusted reference by design, as a diagnostic check of whether the false-consensus pattern persists when evidence is supplied to the same judge panel; we agree the text does not make this best-case assumption explicit. In the revision we will (a) label all grounded results as a **trusted-reference best-case diagnostic**, (b) remove the "collapse-to-zero" and "mechanism-validation" language, and (c) add a limitations paragraph noting that grounded-verification quality depends on reference quality and our numbers should be read as a trusted-reference best case.

---

> > ### Author Response · Authors · 2026-07-15
> > **Author Response (2/2): Cost Analysis, Risk-Framing Clarification, and Revision Plan**
> >
> > *(Part 2 of 2; Part 1 covered Changes 2 and 4.)*
> >
> > **Change 3: JuryProbe vs. Always-Grounded**
> >
> > As requested, we compute dollars per check from current OpenRouter prices and provider-reported token counts for the complete prompt sent to each judge, including references in grounded calls; reference acquisition/retrieval is excluded. A judge call uses roughly 100 input tokens and 2 output tokens; a three-judge panel evaluation costs about 0.000011 USD in either mode. Counting all deployment-time judge calls on each held-out split, after calibration, JuryProbe spends slightly more on LLM calls than Always-Grounded in the high-risk regime because the reference-free panel runs on every claim plus grounded re-checks on accepts (about 0.005 USD per split versus 0.0034 USD). The one-time calibration cost is excluded from this per-split deployment comparison and will be reported separately or amortized in the revision. In the no-routing control, LLM-call costs are essentially equal (about 0.0034 USD per split), but the calls differ: JuryProbe uses reference-free judging, while Always-Grounded uses grounded judging for every claim. Thus **LLM-call cost is negligible for both policies and is not where any advantage lies: if a trusted reference were freely available for every claim, Always-Grounded would dominate the evaluated JuryProbe policy.**
> >
> > The deployment comparison therefore clarifies the role of JuryProbe rather than reducing it to an LLM-call cost optimization. JuryProbe addresses a different question: whether universal grounding is necessary, or whether reference use can be conditioned on an empirical diagnosis of panel error dependence.
> >
> > Under the evaluated policy, JuryProbe requires trusted references for 49.6% (Number) and 62.1% (Entity) fewer deployment claims, and no deployment-time references in the negative control, where its false-accept rate is 0.013 versus 0.000 for Always-Grounded. We also report the coverage tradeoff transparently: the one-sided accept-protection policy accepts 55–64% of clean claims, compared with 100% under Always-Grounded. The revision will therefore present a conditional operating regime rather than claim unconditional dominance: JuryProbe is relevant when false accepts are costly, references are available on demand but expensive to acquire universally, and the system can trade some true-accept coverage for fewer trusted-reference acquisitions. A fuller parameterized utility analysis will make this boundary explicit.
> >
> > **Change 1: Risk guarantee — downweighting the framing**
> >
> > The submitted method does not provide a distribution-free guarantee, and we agree the term "guardrail" and the comparison to [1, 2] invite that stronger reading. In the revision we will (a) state explicitly what is and is not guaranteed, (b) reframe JuryProbe as an empirical **consensus-risk diagnostic with calibration-based routing**, and (c) reposition [1, 2] as complementary: their guarantees target judge–human agreement and single-model selective prediction, whereas our object is the panel's error-dependence structure.
> >
> > Across the evaluated threshold sweeps, Number and Entity remain flagged in all splits, whereas the negative control remains unflagged. A correlation-only rule would flag 2/10 control splits, whereas the full conjunction flags none, illustrating why lift is included. This is an empirical protocol, not a user-specifiable error-rate target; the permutation test does not bound deployment error.
> >
> > **Scope precision.** One clarification: "all 10 splits" applies to Number and Entity only (Sec. 5.3). Attribute satisfies the high-risk criteria in the full-family analysis (Table 1) as a replication family, and is excluded from the trusted-reference diagnostic and routing experiments because its absolute false-consensus rate is too low for stable paired comparison. We will make this scope clearer.
> >
> > **Revised contribution.** The thresholds are an operational instrument, not the contribution. The central contribution is the empirical link between calibration-time panel error dependence and deployment-time unanimous false-consensus risk. Correlated judge errors can produce excess unanimous false consensus that disagreement-based escalation cannot detect by construction. Under one unchanged calibration rule, the diagnostic selects grounded routing for reference-free accepts in Number and Entity and the no-routing branch for the negative control. In the evaluated settings, this provides empirical evidence that calibration-time dependence can inform whether panel agreement is routed to additional verification, with selective grounding as one operational consequence. The claim is empirical rather than a formal risk guarantee, and JuryProbe is not intended as a universal replacement for Always-Grounded.
> >
> > [1] Jung et al., "Trust or Escalate: LLM Judges with Provable Guarantees for Human Agreement," ICLR 2025.
> >
> > [2] Geifman & El-Yaniv, "Selective Classification for Deep Neural Networks," NeurIPS 2017.

---

### Review · Reviewer_37Ua · 2026-07-17

**Summary Of Contributions:**

Reference-free LLM judge panels are trusted when they agree, but agreement is only evidence if judge errors are independent. If judges share false-negative blind spots, a corrupted claim can be unanimously accepted. This paper proposes a method to calibrate this scenario.

## Strengths

- Clear motivation and target: unanimous false accepts are structurally invisible to disagreement- or uncertainty-based escalation, and the paired same-judges experiment (grounding collapse) makes the diagnosis credible rather than just plausible.
- Rigor experiment setting: frozen data and thresholds fixed before evaluation, calibration/deployment splits over 10 seeds, a budget-matched random baseline.

## Weaknesses

- The evaluation setting feels quite narrow to me. Everything is built on short FEVER claims with synthetic corruptions that flip a number or swap an entity, and the examples in Table 6 show just how simple these edits are. Since consensus risk is estimated per corruption family, and the authors themselves note in Section 6.1 that these estimates may not transfer to unseen corruption types, it is unclear whether any of this carries over to natural hallucinations, long-form outputs, or error patterns not seen during calibration, which is arguably where such a guardrail would matter most.

- The method assumes a clean and correct reference is available at verification time. In the experiments the reference is essentially the gold FEVER evidence, so grounded verification runs under ideal conditions. As far as I can tell, a realistic deployment would depend on retrieval, where the reference can be noisy, incomplete, or wrong, and the paper explicitly leaves that interaction unstudied (Section 6.1). This makes the headline 0.000 false accept rate in Table 3 read more like a best case than a deployment number.

- If I understand the protocol correctly, the policy has two branches, trusting the panel when h=0 and routing accepts when h=1, but only the second branch is ever exercised on reference-free data. In both Number and Entity the panel is flagged high-risk in all 10 out of 10 splits (Section 5.3), so the paper never shows a reference-free panel that is correctly identified as low-risk and safely trusted.

- The grounding collapse result feels close to guaranteed by construction here. Each corruption directly contradicts a single fact in the reference, for example a release date changed from July 12 to July 14 (Table 6), and the grounded prompt simply asks whether the claim is fully consistent with that reference (Appendix D.1). At that point the task reduces to text matching that even small judges can hardly fail, so the drop to zero false consensus in Table 2 is not very surprising.

**Audience:**

Yes

**Audience Explanation:**

Judge panels are now a standard component of LLM evaluation pipelines, and the specific observation that unanimous acceptance can itself be the dangerous event, invisible by construction to disagreement-based or uncertainty-based escalation, is something anyone building or auditing such pipelines would want to know.

**Broader Impact Concerns:**

There is no ethics concern about this submission.

**Claims And Evidence:**

No

**Claims Explanation:**

The first claim is that JuryProbe decides when reference-free panel agreement should be trusted and when it should be routed to grounding (abstract, Section 1). If I understand the protocol correctly, that decision has two branches, trusting the panel when h=0 and routing accepts when h=1, but only the second branch is ever exercised on reference-free data. In both Number and Entity the panel is flagged high-risk in all 10 out of 10 splits (Section 5.3), so the paper never shows a reference-free panel that is correctly identified as low-risk and safely trusted. The grounded specificity check in Appendix B (0/10) is computed on grounded outputs, so as far as I can tell it does not test that branch either. What the evidence establishes is that routing accepts to grounding helps when a panel is high-risk. It does not establish that the estimator can discriminate between panels that should and should not be trusted, which is the selling point that separates JuryProbe from a simple rule like "always ground accepts from small reference-free panels."

The second claim is the mechanism validation, that consensus risk collapses under grounding and therefore arises from judging without evidence (contribution 2, Section 5.2). This result feels close to guaranteed by construction. Every corruption directly contradicts a single fact in the reference, for example a release date changed from July 12 to July 14 (Table 6), and the grounded prompt simply asks whether the claim is fully consistent with that reference (Appendix D.1). At that point the task reduces to a text match that even small judges can hardly fail, so the zero grounded false consensus in Table 2 is not very surprising and, from my understanding, mostly shows that the reference-augmented protocol works on easy contradictions. The authors partially concede this in Section 6.1, but the paper still presents it as mechanism-level evidence for the broader diagnosis.

**Requested Changes:**

Please refer to the weakness part.

---

### Review · Reviewer_M63q · 2026-07-20

**Summary Of Contributions:**

This paper introduces JuryProbe, a guardrail for reference-free LLM judge panels. It measures correlated false-negative errors using FN-only correlation and false-consensus lift, and routes high-risk accept decisions to grounded verification.

Experiments on FEVER-derived Number and Entity corruptions show that reference-free judges can unanimously accept false claims due to shared blind spots. Providing trusted references to the same judges eliminates false consensus in the evaluated settings. JuryProbe-Routed achieves zero false accepts with fewer verifier calls than Always Grounded.

The paper identifies an important failure mode, clearly distinguishes agreement from independent evidence, and provides useful comparisons with disagreement-based and random routing.

**Audience:**

Yes

**Audience Explanation:**

The finding that unanimous judge agreement may result from shared blind spots is relevant to researchers working on LLM evaluation, factuality, hallucination detection, judge panels, and selective verification.

The paired comparison using the same judges with and without references is also informative for designing reliable and cost-aware evaluation systems.

**Broader Impact Concerns:**

No major ethical concern prevents publication.

However, the paper should note that:
- A low-risk classification may not transfer to new domains, models, languages, or corruption types.
- Trusted references may be incomplete, incorrect, outdated, or manipulated.
- Automated factuality judging should not replace expert review in high-stakes applications without domain-specific validation.
- Model, prompt, retrieval, or data changes may require recalibration.

**Claims And Evidence:**

No

**Claims Explanation:**

The experiments convincingly support the claims that reference-free judges exhibit correlated false negatives, that disagreement-based routing cannot catch unanimous false accepts, and that grounding improves reliability.

However, they do not establish that the improvement is specifically caused by consensus-risk estimation. Since every evaluated split is labeled high-risk, the risk estimate never changes the policy. A baseline that always grounds reference-free accepts would behave identically.

The authors should either evaluate low-risk settings where JuryProbe avoids unnecessary grounding, or narrow the claims to focus on correlated false consensus and accept-conditioned grounding.

**Requested Changes:**

Critical
- Add a baseline that grounds every reference-free majority accept without using the risk estimator.
- Evaluate settings where JuryProbe produces both high-risk and low-risk classifications, showing when grounding can safely be avoided.
- Otherwise, revise claims that the improvement comes specifically from “routing on measured consensus risk.”
- Clarify how the correlation and lift thresholds were selected and whether evaluation data influenced them.


Non-critical
- Report event counts and uncertainty intervals for correlation, false-consensus rate, and lift.
- Include calibration and reference-acquisition costs in the efficiency discussion.
- Provide fuller reproducibility details, including prompts, model settings, corruption construction, audits, and seeds.
- Limit the conclusions more clearly to the evaluated controlled factuality setting.